# Acetylsalicylic Acid and Salicylic Acid Inhibit SARS-CoV-2 Replication in Precision-Cut Lung Slices

**DOI:** 10.3390/vaccines10101619

**Published:** 2022-09-27

**Authors:** Nina Geiger, Eva-Maria König, Heike Oberwinkler, Valeria Roll, Viktoria Diesendorf, Sofie Fähr, Helena Obernolte, Katherina Sewald, Sabine Wronski, Maria Steinke, Jochen Bodem

**Affiliations:** 1Institute for Virology and Immunobiology, Julius-Maximilians-University of Würzburg, Versbacher Strasse 7, 97078 Wuerzburg, Germany; 2University Hospital Wuerzburg, 97080 Wuerzburg, Germany; 3Preclinical Pharmacology and In-Vitro Toxicology, Fraunhofer Institute for Toxicology and Experimental Medicine ITEM, 30625 Hannover, Germany; 4Fraunhofer-Translationszentrum für Regenerative Therapien TLZ-RT, Fraunhofer Institute for Silicate Research ISC, Röntgenring 11, 97070 Wuerzburg, Germany

**Keywords:** acetylsalicylic acid, salicylic acid, antiviral activity, aspirin, SARS-CoV-2, precision-cut lung slices

## Abstract

Aspirin, with its active compound acetylsalicylic acid (ASA), shows antiviral activity against rhino- and influenza viruses at high concentrations. We sought to investigate whether ASA and its metabolite salicylic acid (SA) inhibit SARS-CoV-2 since it might use similar pathways to influenza viruses. The compound-treated cells were infected with SARS-CoV-2. Viral replication was analysed by RTqPCR. The compounds suppressed SARS-CoV-2 replication in cell culture cells and a patient-near replication system using human precision-cut lung slices by two orders of magnitude. While the compounds did not interfere with viral entry, it led to lower viral RNA expression after 24 h, indicating that post-entry pathways were inhibited by the compounds.

## 1. Introduction

Starting in December 2019, SARS-CoV-2 became a pandemic threat with more than 580 million cases worldwide and more than 6.35 million deaths until the summer of 2022, with still rising numbers due to new, more efficiently transmitted variants of SARS-CoV-2 and lower prevention measures [1]. In contrast to previous years, this combination further increased infection rates in Western countries even during the summer [1]. The high infection rates compensated in Germany in parts the effects of the introduction of efficient vaccines leading to death rates comparable to April 2020 at the beginning of the pandemic, when no vaccine had been available. Unfortunately, efficient direct-acting therapies are still widely unavailable and have severe adverse side effects, such as the potential teratogenicity of molnupiravir [2].

The repurposing of approved medications offers a fast option besides developing new direct-acting antiviral drugs; since these compounds are available, dosing and adverse side effects are already known. Furthermore, established compounds are often more cost-effective than newly developed antivirals, such as sofosbuvir. Several approaches for off-label use of approved drugs against SARS-CoV-2, such as remdesivir [3,4], fluoxetine [5] and fluvoxamine [6], have been published. Some were incorporated into the national treatment guidelines, but others, for instance, chloroquine, failed in patients due to inadequate preclinical test systems, such as the usage of Vero E6 cells [7]. Advanced tissue systems such as 3D models of lung or upper airway tissues or human precision-cut lung slices (PCLS) have rarely been used for antiviral assays [5]. However, compounds analysed on human PCLS, such as fluoxetine, were beneficial in clinical studies. Moreover, the analyses of the antiviral mechanisms of these compounds led to the identification of new side activities of these drugs, such as the inhibition of the acidic ceramidase by fluoxetine [8].

Regarding drug-repurposing, aspirin, a compound used for more than one century, was shown to be effective against influenza A and rhinoviruses [9]. ASA was found to have anti-inflammatory and anti-platelet effects in a dose-dependent way (reviewed in [10]). Furthermore, clinical studies with small patient numbers indicated that treatment with aspirin lowered both the 30- and 60-day mortality in the aspirin group significantly [11]. A more extensive retrospective cohort study found that patients receiving aspirin had less oxygen support at admission than patients not receiving aspirin [12]. However, studies showing a direct antiviral effect against SARS-CoV-2 have not been published so far.

The analyses of the viral replication cycles are a prerequisite for developing and improving new antivirals and understanding where “off-label” drugs might interfere with viral replication. SARS-CoV-2 is a plus-stranded, enveloped RNA virus. It uses the spike protein to enter the cells via angiotensin-converting enzyme 2 (ACE2), which is abundantly expressed in alveolar and heart tissue [13]. The virus enters the cell via cell type-specific pH-dependent endocytosis or by direct fusion with the cytoplasmic membrane depending on the concentration of the cellular TMPRSS2 proteases [14]. Inhibition of viral entry will result in lower RNA amounts early after infection. The incoming viral RNA is translated into viral proteins and expressed by the viral RNA polymerase [15]. Thus, inhibition of viral replication steps after entry until RNA expression will result in a lower increase of RNA amounts in the cells. Here, we analyse whether acetylsalicylic acid or its metabolite salicylic acid inhibits SARS-CoV-2 replication in cell culture and human PCLS. Furthermore, we provide evidence that the compounds do not interfere with viral entry but with replication steps before or during gene expression.

## 2. Materials and Methods

### 2.1. Cells Lines, Compounds and Viruses

A previously described, patient-derived, and completely sequenced SARS-CoV-2 isolate was used for infection experiments [5,8,16]. Vero, Huh-7, and A549-ACE2 cells were cultured in DMEM medium (Life Technologies, Darmstadt, Germany) supplemented with 10% foetal calf serum (Merck-Sigma Aldrich, Taufkirchen, Germany). Acetylsalicylic and salicylic acids were purchased from Merck-Sigma Aldrich at the standards of the American Chemical Society.

### 2.2. Cellular Toxicity 

Cell toxicity was determined by analysing cell growth. The cells were seeded into an optical 96-well plated and counted on the following day with a PerkinElmer Ensight device. The compounds were added, the cells were incubated for 72 h, and cell numbers were determined again. The growth was determined by dividing the cell number per well after 72 h by the cell number before adding the compounds and after 72 h of incubation. The cell growth was determined by dividing the cell numbers per well. The assay was performed in six independent samples, and the standard deviation was calculated.

### 2.3. RNA Purification and Determination of Viral Genome Copies

Viral replication was quantified by determining the genome copy number by RTqPCR 72 h after infection from the cell culture supernatants. The cells were incubated with the compounds and subsequently infected with SARS-CoV-2, as described before [5]. The medium was exchanged after 24 h with the medium containing the compounds. After 72 h, 250 µL of the medium was collected, and viral genomes were purified with the MagNA Pure 24 system (Roche, Mannheim, Germany). SARS-CoV-2 RNA genomes were quantified with the dual-target SARS-CoV-2 RdRP RTqPCR assay kit with RNA Process Control PCR Kit (Roche, Mannheim, Germany) with a LightCycler 480 II (Roche, Mannheim, Germany). The genome copy numbers were determined with the LightCycler software version 1.5 using the provided positive control as standard.

Similarly, we quantified viral RNAs from cells by isolating total RNAs from cells. The RNAs were prepared according to the manufacturer’s instructions (Omega, VWR, Darmstadt, Germany). An amount of 150 ng of total RNA was used in RTqPCR reactions to quantify SARS-CoV-2 RNA, as described above. Results were normalised on GAPDH expression by RTqPCR using the primers (5′-ACAACGAATTTGGCTACAGC-3′, 5′-AGTGAGGGTCTCTCTCTTCC-3′) and the hydrolysation probe ([Cy5]-ACCACCAGCCCCAGCAAGAGCACAA-[BHQ]).

### 2.4. Human Precision-Cut Lung Slices

Infection of human precision-cut lung slices (PCLS) and determination of viral infections were performed as described before [5,17]. After the transport, human PCLS were incubated for 1 h at 37 °C in a DMEM/F12 medium (Life Technologies, Darmstadt, Germany) supplemented with 1% penicillin/streptomycin (Lonza, Verviers, Belgium) and separated on a 48-well dish. The compounds were added, and the PCLS were infected with SARS-CoV-2 with a high MOI of approximately 10. The resulting viral infectivity was determined by infecting Vero cells with 100 µL of the viral supernatants for two days and analysing viral genome amounts in the cell culture supernatant by RTqPCR.

## 3. Results and Discussion

Cell culture-based infection experiments were conducted to determine the potential effects of ASA and SA on SARS-CoV-2 replication. First, the cytotoxicity of the compounds was analysed, and secondly, the impact on viral replication rates was measured. A549-ACE2, Huh-7, and Vero cells were incubated with ASA or SA for three days to determine cytotoxicity, and the relative cell growth was analysed by automatic cell counting as described before [5]. Concentrations of 3 mM SA or ASA did not influence cell growth.

Since it has been shown that the antiviral effect of a defined compound on SARS-CoV-2 replication depends on the applied test system [18], the influence of ASA and SA on viral replication in Vero, A549-ACE2, and Huh-7 cells was compared. We decided to use Vero and Huh-7 cells, the first cell lines frequently used in SARS-CoV-2 infection experiments. Thus, our results should be comparable to previous studies. At the same time, the adenocarcinoma human alveolar basal epithelial A549 cell line was chosen since it represents an infection model near the relevant tissue.

The cells were pre-incubated with the compounds at 1.5 and 3 mM and infected with SARS-CoV-2 in triplicates (Figure 1). All infections were repeated twice. Cellular supernatants were collected two days after infection. Viral RNAs were extracted using a MagNA Pure 24 system. SARS-CoV-2 RNA genomes were quantified with the dual-target SARS-CoV-2 RdRP RTqPCR assay kit with RNA Process Control PCR Kit with a LightCycler 480 II (Figure 1). ASA and SA significantly inhibited viral replication in a cell type-specific way by approximately half (Huh-7) or two (Vero cells and A459-ACE2) orders of magnitude, respectively (Figure 1A–C). This reduction is above the previously reported suppression of the respiratory syncytial, influenza and rhinoviruses. It indicates that ASA and SA interfere with viral replication [9], indicating that the compounds might inhibit an essential SARS-CoV-2 pathway. Furthermore, we show that the inhibition of SARS-CoV-2 replication is independent of the cell lines used and that both compounds suppress viral replication even in the most relevant A549 cells.

However, recent studies with antiviral drugs against SARS-CoV-2 have revealed that the translation from a cell culture-based system, such as VeroE6 cells, might result in inactive patient therapies. Chloroquine, for example, shows antiviral activity in these cells but not in Calu3 cells, human epithelial tissue models, macaques or in patients [7,18]. Therefore, we decided to determine the effects of ASA and SA in human PCLS, as described before [5]. The PCLS were incubated for 1 h at 37 °C, incubated with the compounds, and subsequently infected with SARS-CoV-2 at a high MOI of 10. After 3 days, 10 µL of the cell culture supernatants were used to infect Vero cells to determine the infectious viruses. Cellular supernatants were collected, RNAs isolated, and viral genomes quantified after 3 days by RTqPCR (Figure 1D). ASA inhibited viral replication by approximately one order of magnitude, while the treatment with SA reduced viral loads by more than two. The treatment with the compounds reduces viral replication in all cell lines analysed and in PCLS, indicating that they are promising candidates to be evaluated in antiviral therapy.

Next, we sought to determine whether ASA and SA inhibit the entry of SARS-CoV-2. A549-ACE2 cells were incubated with the substances, and viral RNA expression was determined 6 h and 24 h after infection. All infections were performed in independent triplicates. After one hour, the medium was exchanged to remove the virus from the supernatant. After 6 h or 24 h, the cells were washed with ice-cold PBS, and the RNAs were prepared. Similar RNA amounts were used in RTqPCR reactions to quantify SARS-CoV-2 RNA as described above. The results were normalised on GAPDH by RTqPCR. After 6 h of infection, we observed similar SARS-CoV-2 RNA amounts in all lysates (Figure 2A), providing evidence that equal amounts of the virus could enter the cells despite the treatment. Thus, we conclude that the viral entry process is not targeted in SARS-CoV-2 infections. The low amount of viral RNAs at 6 h indicates that this time-point is before substantial viral gene expression.

Similar experiments were performed to analyse the infection after 24 h in the presence of both compounds (Figure 2B). We observed a reduction of viral RNA amounts after 24 h with ASA at concentrations higher than 1.5 mM. In comparison, SA reduced viral RNAs at lower concentrations, like the above results. These results indicate that the compound inhibits replication steps after viral entry but before or during gene expression.

It has been shown that SA and ASA influence influenza virus replication by blocking NF-κB activation, which is essential for synthesising viral genomic RNAs [19,20,21]. Furthermore, SARS-CoV-2 activates NF-κB early in infection, indicating that inhibition of the NF-κB pathway could suppress virus replication [22]. Thus, we collected evidence that the observed reduction of SARS-CoV-2 replication by ASA and SA could be attributed to the inhibition of NF-κB.

## 4. Conclusions

In summary, ASA and SA act effectively against SARS-CoV-2 in different animal- and human-derived cell lines and in precision-cut, patient-derived lung slices. We provide evidence that both compounds might also decrease viral loads in patients and might be further analysed in prospective clinical trials.

## Figures and Tables

**Figure 1 vaccines-10-01619-f001:**
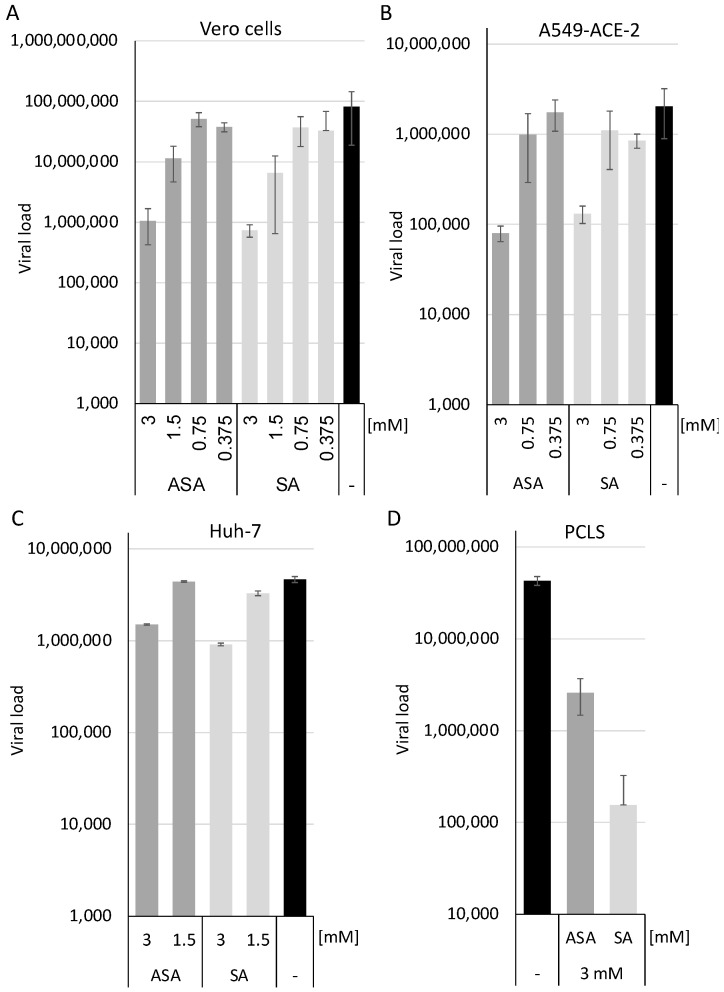
SA and ASA inhibit viral replication in Vero, A549-ACE2, Huh-7 cells and human precision-cut lung slices. Vero (**A**), A549-ACE2 (**B**) and Huh-7 (**C**) cells, as well as PCLS (**D**), were treated with the compounds and infected with SARS-CoV-2. Viral loads were determined 3 days after infection. Bars represent the mean, and error bars the standard deviation.

**Figure 2 vaccines-10-01619-f002:**
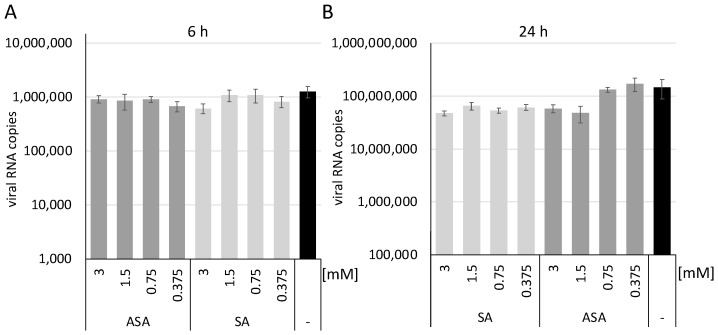
SA and ASA do not interfere with viral entry but with replication steps before or during gene expression. Total RNAs of SARS-CoV-2 infected cells were isolated after 6 (**A**) and 24 h (**B**), and SARS-CoV-2 RNA was quantified by RTqPCR. Bars represent the mean, and error bars the standard deviation.

## Data Availability

Not applicable.

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
