# Peer review of "Acetylsalicylic Acid and Salicylic Acid Inhibit SARS-CoV-2 Replication in Precision-Cut Lung Slices"

_vaccines, 2022, doi:10.3390/vaccines10101619_

Round 1

Reviewer 1 Report

The paper is of high interest, concerning the abundant use of acetylo salicic acid in population. The methodology is very interesting but it needs significant improvement: authors describe the use of cells lines in Figures, but there is no relevant information in materials and methods. The rationale for choosing the particular cell lines is also missing.

Author Response

>The paper is of high interest, concerning the abundant use of acetylo salicic >acid in population. The methodology is very interesting but it needs significant >improvement: authors describe the use of cells lines in Figures, but there is no >relevant information in materials and methods.

We added this information to materials and methods (lines 74-79).

>The rationale for choosing the particular cell lines is also missing.

We added this information to the results (lines 124-128).

Reviewer 2 Report

This paper talks about ASA and SA inhibiting SARS-CoV-2 replication. Since COVID is still a problem worldwide, drug development against COVID infection is very important. So, the topic of this article has strong significance. However, there are still things need to be considered. 

1.     Please check your references. For instance, the last paragraph of your introduction, you need to add citations. Usually, we add citations after each solid statements. 

2.     Also, it would be better if the introduction is enriched with more information. For instance, when you talk about the virus structure, like the spike protein, or when you mention the list of drugs, you can talk more on what is known of the antivirals binding sites, structurally. When readers go through the article, they may be curious on how, a compound, or a drug like aspirin, related to the virus itself, what’s the pharmacology. and what’s the mechanism behind it. Please try to build bridges between the information you provide to readers. Currently, the transitions between paragraphs are slightly rigid. 

3.     Things make sense in this paper. However, instead of pre-incubate cells with the compounds, have you tried to do things in the opposite way: incubate the virus with compounds and then test their infections on cells?

4.     Also, you mentioned about other respiratory viruses, like Influenza and Rhino. Then more discussions should be done, such as… say an easy one: any structural data available showing their binding sites? Any similarities or differences of ASA and SA interfering the viral replication in different viruses? 

5.     Go back to the compound, or related drugs like aspirin. More discussion is needed, like what is the pros or cons, right, using it against COVID? Any potential side effects?

Author Response

1 Please check your references. For instance, the last paragraph of your introduction, you need to add citations. Usually, we add citations after each solid statements. 

We added references to the introduction.  

  1. Also, it would be better if the introduction is enriched with more information. For instance, when you talk about the virus structure, like the spike protein, or when you mention the list of drugs, you can talk more on what is known of the antivirals binding sites, structurally. When readers go through the article, they may be curious on how, a compound, or a drug like aspirin, related to the virus itself, what’s the pharmacology. and what’s the mechanism behind it. Please try to build bridges between the information you provide to readers. Currently, the transitions between paragraphs are slightly rigid. 

We added information on aspirin as a new paragraph (lines 49-56) with references.

We think that aspirin might work by blocking NF-kappa B activation, and we added this information in a new paragraph at the end of the manuscript.

  1. Things make sense in this paper. However, instead of pre-incubate cells with the compounds, have you tried to do things in the opposite way: incubate the virus with compounds and then test their infections on cells?

We disagree with the reviewer since we show in the paper that the entry is not targeted by the compounds (Figure 2), and it is unlikely that the compounds target the virus directly. Despite our concerns, we did the experiment suggested by the reviewer: we pre-incubated 10 µl of our standardized viral supernatant with 90µl of the medium containing the compounds at 1.5 and 3 mM of the compounds for 30 min at room temperature. Then 5 µl of the mixture was used to infect Vero cells, which equals the amount of virus used in all other experiments. After 72 h cell culture supernatants were collected, and viral RNAs were extracted. We determined the viral genome copy number as described in the manuscript.    

Image is provide in the Word file 

We did not observe significant differences in ASA or SA pretreated samples compared to the medium control.       

  1. Also, you mentioned about other respiratory viruses, like Influenza and Rhino. Then more discussions should be done, such as… say an easy one: any structural data available showing their binding sites? Any similarities or differences of ASA and SA interfering the viral replication in different viruses? 

We added this information to a new paragraph at the end of the manuscript.

  1. Go back to the compound or related drugs like aspirin. More discussion is needed, like what is the pros or cons, right, using it against COVID? Any potential side effects?

We added a paragraph to the end of the manuscript and modified the conclusions.

Round 2

Reviewer 1 Report

No further comments

Reviewer 2 Report

The content of the paper has been enriched in a proper way, indicating that the manuscript is ready for publication. The overall significance of the study is high, and hopefully people in the field will learn/be benefited from this work.